# Glucose Transport in *Escherichia coli*: From Basics to Transport Engineering

**DOI:** 10.3390/microorganisms11061588

**Published:** 2023-06-15

**Authors:** Ofelia E. Carreón-Rodríguez, Guillermo Gosset, Adelfo Escalante, Francisco Bolívar

**Affiliations:** Departamento de Ingeniería Celular y Biocatálisis, Instituto de Biotecnología, Universidad Nacional Autónoma de México, Av. Universidad 2001, Cuernavaca 62210, Morelos, Mexico; efelio2000@gmail.com (O.E.C.-R.); guillermo.gosset@ibt.unam.mx (G.G.)

**Keywords:** *Escherichia coli*, carbohydrate transport, PTS, ABC transporter, MFS transporter, genetic regulation, carbohydrate transport engineering

## Abstract

*Escherichia coli* is the best-known model for the biotechnological production of many biotechnological products, including housekeeping and heterologous primary and secondary metabolites and recombinant proteins, and is an efficient biofactory model to produce biofuels to nanomaterials. Glucose is the primary substrate used as the carbon source for laboratory and industrial cultivation of *E. coli* for production purposes. Efficient growth and associated production and yield of desired products depend on the efficient sugar transport capabilities, sugar catabolism through the central carbon catabolism, and the efficient carbon flux through specific biosynthetic pathways. The genome of *E. coli* MG1655 is 4,641,642 bp, corresponding to 4702 genes encoding 4328 proteins. The EcoCyc database describes 532 transport reactions, 480 transporters, and 97 proteins involved in sugar transport. Nevertheless, due to the high number of sugar transporters, *E. coli* uses preferentially few systems to grow in glucose as the sole carbon source. *E. coli* nonspecifically transports glucose from the extracellular medium into the periplasmic space through the outer membrane porins. Once in periplasmic space, glucose is transported into the cytoplasm by several systems, including the phosphoenolpyruvate-dependent phosphotransferase system (PTS), the ATP-dependent cassette (ABC) transporters, and the major facilitator (MFS) superfamily proton symporters. In this contribution, we review the structures and mechanisms of the *E. coli* central glucose transport systems, including the regulatory circuits recruiting the specific use of these transport systems under specific growing conditions. Finally, we describe several successful examples of transport engineering, including introducing heterologous and non-sugar transport systems for producing several valuable metabolites.

## 1. Introduction

Glucose is the essential carbon source for growing and cultivating heterotrophic bacteria, such as *Escherichia coli*, for laboratory and production purposes. This sugar is the primary carbon and energy source for large-scale biotechnological processes and provides faster and optimum growth compared with other carbon sources. *E. coli* preferentially uses glucose in the presence of sugar mixtures, preventing using other carbon sources. Several transcriptional and post-transcriptional regulatory mechanisms control the preferential use of glucose over other sugars. The transcriptional control mechanism known as carbon catabolite repression (CCR) prevents the expression of more than 180 genes (including transport and catabolic genes) and the inducer exclusion mechanism, where the uptake or synthesis of an inducer molecule of a sugar catabolic operon is prevented [1,2,3,4,5,6,7,8,9,10].

The genome size of *E. coli* strain K-12 MG1655 is 4,641,652 bp, corresponding to 4702 genes encoding 4328 proteins, 228 RNA genes, and 146 pseudogenes. Transport comprises 532 reactions, including 480 transporters (EcoCyc database https://biocyc.org/ECOLI/organism-summary, accessed on 1 May 2023) [11]. Among them, 97 proteins are involved in sugar transport (Table 1). Additionally, numerous transporters with overlapping sugar specificities for monosaccharides increase the potential capability to transport glucose [6], indicating the extraordinary capability and plasticity of transporting and growing glucose as a carbon source. In contrast to the higher sugar transport systems included in *E. coli* K12, according to the BioCyc database [12], other organisms such as *Salmonella enterica* serovar Typhimurium str LT2 possess just three glucose transmembrane transporters, *Listeria monocytogenes* 10403S do not report any hexose transporter, and *Pseudomonas aeruginosa* PA01 reports only two hexose importers [11].

According to the high capability to transport glucose, wild-type strains of *E. coli* can grow efficiently in minimal broth, such as M9 broth supplemented with glucose as the sole carbon source, achieving higher specific growth rates (μ), e.g., *E. coli* K12 shows a μ = 0.57 h^−1^ [13], strain MG1655, μ = 0.92 h^−1^, and the derivative strain JM101, μ = 0.7 h^−1^ [5]. The transport and breakdown of imported glucose through the glycolytic pathway supplies at least 12 biosynthetic precursors necessary for the biosynthesis of all the structural blocks of the cell from this carbon source [11].

The outer and inner membrane in *E. coli* imposes two different processes for glucose transport from the extracellular medium into the cytoplasm (Figure 1). The outer membrane acts as a molecular sieve to pass diverse hydrophilic molecules such as glucose. Extracellular solutes enter by diffusion through the inner channel of the outer membrane porins (OMP) into the periplasmic space in a non-selective process, limited only by the cutoff size of the OMP inner channel and the physicochemical properties of the solutes. However, some specificity is observed in some OMPs, such as LamB [14,15]. Different transporters mediate the import of periplasmic glucose into the cytoplasm against a gradient concentration mechanism, comprising (i) the phosphoenolpyruvate (PEP)/glucose Phosphotransferase-driven Group Translocators (PTS) systems, (ii) the primary active glucose transporters of the ATP-Binding Cassette (ABC) superfamily, specifically, ATP-dependent transporters, and (iii) the secondary active solute (glucose)/cation symporters members of the Major Facilitator Superfamily (MFS), utilizing H^+^ proton gradients maintained by the ATPases system (Table 1) [6,11,16,17]. In this contribution, we review the characteristics and mechanisms of the abovementioned glucose transporter systems in *E. coli*, the regulatory circuits recruiting the specific or concomitant use of these transport systems under specific growing conditions (e.g., switching from glucose-rich to glucose-limited conditions), and the cross-taking interactions between several transporters resulting in the unspecific glucose transport. Finally, we describe several examples of transporter engineering, including introducing heterologous and non-sugar transport systems to produce several valuable metabolites efficiently. 

## 2. The Cellular Membrane in *Escherichia coli* and Solute Transport by Outer Membrane Porins (OMP)

*E. coli* possesses a two-membrane envelopment system. The outer membrane defines the cellular boundary between the extracellular environment with an aqueous space or periplasm containing a thin peptidoglycan layer. The outer membrane is asymmetric, containing lipopolysaccharides (LPS) in the outer layer and phospholipids in the inner or periplasmic layer. LPS acts as an efficient permeability barrier against large hydrophilic and hydrophobic molecules. The inner or cytoplasmic membrane acts as a hydrophilic barrier between the periplasm and the cytoplasm, maintaining the intracellular concentration of small molecules and proteins. Both membranes provide a suitable environment for the anchorage and function of diverse transporter proteins and other components involved in bioenergetic and biosynthetic reactions [14,24,25,26]. Integral OMPs in the outer membrane are composed of amphipathic β-strands, forming a β-barrel structure and forming transmembranal pores that allow passage of solutes across the outer membrane by an energy-independent process. In contrast, the inner membrane proteins span the membrane as α-helices, almost entirely composed of hydrophobic residues [14,26,27,28]. OMPs are specific and non-specific filters preventing the passage of molecules more significant than 600–1000 Da [11,14]. 

It is estimated that there are 10^5^ porins per cell [26], but, in *E. coli* K12, the major non-specific OMPs are the outer membrane porin F (OmpF) and the outer membrane porin C (OmpC). OmpF has a pore diameter slightly larger than OmpC (1.16 nm and 1.08 nm, respectively) [26]. However, despite the slight differences in pore diameter between OmpF and OmpC, the diffusion of solutes is influenced by the solute’s size, electrical charge, and hydrophobicity. Thus, although the molecular weight of a disaccharide (sucrose > 360 Da) and a pentose (e.g., xylose > 150 Da) are below the exclusion limit of OmpF and OmpC (600 Da), the rate of diffusion of the disaccharide is twice that of the pentose [26]. Porins have a relevant role in bacterial physiology. They are induced under growing stress conditions such as high osmolarity or high temperature. However, their abundance modifies the permeability of the outer membrane toward small or larger solutes, e.g., a decrement in the proportion of OmpF and an increment of OmpC modifies the permeability of the cell membrane toward small solutes of 100–200 Da [14,26,29]. Control of the expression of *ompF* is highly regulated, and the increased production under low-temperature and low- osmolarity conditions were reported to benefit *E. coli,* improving the influx of scarce nutrients [15]. The maltose outer membrane channel/phage lambda receptor protein LamB facilitates the diffusion of maltose, other maltodextrins, and glucose across the outer membrane [11,30]. LamB is a member of the *malKlamBmalM* operon in *E. coli* K12 and acts as a porin for the diffusion of low-molecular-weight solutes in the size of monosaccharides [31]. Under high glucose concentration (0.2 mM) in the culture medium, *E. coli* diffuses glucose into the periplasm by OmpF and OmpC [29,32], whereas, under glucose limiting conditions, LamB is induced and diffuses glucose preferentially into the periplasm [31,32,33,34].

## 3. The Cytoplasmic Membrane Transport System: Glucose Transport against a Gradient Concentration Mechanism

### 3.1. The Phosphoenolpyruvate (PEP):Glucose Phosphotransferase System (PTS Glucose)

Environmental bacterial communities require efficient mechanisms to adapt to nutrient (carbohydrates) availability, leading to the modulation of the expression of different transporter systems with different affinities and capabilities, promoting the transport and catabolism of available sugars. In *E. coli,* as in many bacteria, the PTS sugar systems are the primary type of sugar transport; they mediate the concomitant uptake and phosphorylation of specific carbohydrates (PTS sugars) but additionally have a preponderant role both in the control of carbon and nitrogen metabolisms. In the presence of carbohydrate mixtures (PTS and non-PTS sugars), including glucose, PTS is induced, senses, and controls the preferential uptake and catabolism of this sugar over other available non-PTS carbohydrates in the mixture by the Catabolite Control Repression (CCR) and inducer exclusion mechanisms [5,6,17,18,19]. PTS is a primary regulator of other cellular processes such as transcription and antitermination mechanisms, chemotaxis toward PTS sugars, or cell surface rearrangements [19,35,36,37,38].

The basic structure of all the PTS systems is similar among bacterial species comprising two common proteins to them, the cytoplasmic energy-coupling phosphotransferase protein Enzyme I (EI) and the histidine phosphocarrier protein (HPr) [6,17,24,25]. Carbohydrate specificity relies on the EII enzymes, and *E. coli* contains 15 different EII^specific sugar^ complexes (Table 1). EII^Sugar^ enzymes comprise the hydrophobic domains IIA^Sugar^ and IIB^Sugar^ (hydrophilic domains) and the hydrophobic domains IIC^Sugar^ or IICIID^Sugar^. All these domains are responsible for the transport and phosphorylation of the specific carbohydrate [6,10,18]. The common PTS proteins EI and HPr and the membrane transporter subunits/domains are phosphorylated at histidine (P~EI, P~Hpr, respectively), but, in the IIB subunit/domains of the mannose family, the phosphorylated residue is a histidine while a cysteine for all the other families [6,17]. In *E. coli,* the operon *ptsHIcrr* encodes cytoplasmic Hpr, EI, and EIIA^Glc^ enzymes, respectively, whereas the EIICB^Glc^ (PtsG) is encoded by *ptsG* [10,18]. PtsG is oligomeric within the cytoplasmic membrane, consists of a large hydrophobic domain (IIC^Glc^), and contains the glucose binding site and a smaller hydrophilic domain (IIB^Glc^) exposed on the cytoplasmic face of the membrane. EIIB^Glc^ contains the active cysteine residue for phosphotransfer from EIIA^Glc^ to the primary hydroxyl group of the incoming glucose in a convex surface surrounded by hydrophobic residues [11] (Figure 1A). In *E. coli*, the phosphotransference from PEP to the incoming glucose comprises the reversible autophosphorylation of the active dimeric protein EI, the transference of the phosphoryl group to the HPr protein, and the transference of the phosphoryl group to EIIA domain: PEP↔P~EI↔P~HPr↔P~EIIA^Glc^. The phosphoryl group is then transferred to EIIB, P~EIIA^Glc^↔P~EIICB^Glc^, and the final step of the phosphotransference comprises P~EIIB^Glc^ + glucose↔EIIB^Glc^ + glucose-6-phosphate (G6P). K_m_ of EIIB^Glc^ = 10–20 μM depending on the analyzed strain and growth conditions (Figure 1A) [7,17,18,39,40,41]. *E. coli* internalizes glucose through two other PTS systems: the mannose (EIIAB^Man^) and the *N-*acetylglucosamine-specific PTS enzyme II (EIICBA^Nag^). EIIAB^Man^ has shown the capability to transport and phosphorylate glucose as the sole carbon source under aerobic conditions and has a higher affinity to glucose (*K_m_* = 15 μM). EIICBA^Nag^ is homologous to PtsG but exhibits marginal glucose transport capacity [6,7,11,20]. 

PEP has the highest phosphate bond energy in living organisms (−61.0 KJ/mol). The standard free energy of phosphotransference from PEP to incoming sugar is −48 KJ/mol, where 80% is released in the last stage of the process from P~Cys-EIIB^sugar^ to the primary OH- of the incoming sugar [18]. The intracellular concentration of PEP was estimated between 0.5 and 3 mM and was estimated at 50 μM for G6P [18]. The PEP/pyruvate boundaries were determined in 2800/900 μM; for glucose/G6P, it was 500/50 μM, and the concentration of PTS proteins calculated on their molecular masses were established in [EI^Glc^]_Tot_, [HPr]_Tot_ = 5, 50, respectively, and [IIA^Glc^]_Tot_, [IICB^Glc^]_Tot_ = 40, 10, respectively [41]. Based on the in vivo uptake of the glucose analog α-D-methyl-glucoside, an estimated concentration of EIICB^Glc^ of ~10 μM and a cellular volume of ~2.5 μL per mg dry weight, Franke et al. (2002) calculated that 37 molecules of glucose are imported and phosphorylated by one molecule of IICB^Glc^ per second [41]. Two molecules of PEP result from the catabolism of one molecule of G6P imported by PTS. One PEP acts as a phosphoryl donor for the incoming glucose, whereas the second PEP can be channeled as a biosynthetic precursor [18].

#### 3.1.1. Control of Carbon Metabolism in *E. coli* by PTS Glucose by CCR and Inducer Exclusion Mechanisms: The Role of EIICD^Glc^ and EIIA^Glc^

When *E. coli* grows in glucose in mixtures with non-PTS sugars such as those derived from lignocellulosic hydrolysates containing a glucose–xylose mixture or in glucose–lactose or glucose–glycerol mixtures, PTS controls the preferential transport and consumption of glucose or other PTS sugars. In contrast, the mixtures’ secondary or alternative carbon sources are not transported and catabolized by CCR and inducer exclusion mechanisms. The preferential transport and catabolism of glucose were proposed because this sugar enters directly into glycolysis [6]. In contrast, non-PTS sugars, such as lactose, are transported into the cell, intracellularly hydrolyzed to yield the glucose + galactose moieties which are then transported out of the cell, followed by glucose reentry by PTS:glucose prior it enters to the glycolysis [11,42].

CCR involves the cyclic AMP (cAMP), the second messenger for intracellular signal transduction, the DNA-binding transcriptional dual regulator CRP, the membrane-bounded protein adenylate cyclase (Cya), and the EIIA^Glc^ component of PTS [11] (Figure 1B). In the presence of glucose and other PTS sugars, the PTS proteins are found in the dephosphorylated form, whereas bacterial growth in non-PTS carbohydrates (in the absence of glucose) and the presence of PEP or growing in poor carbon sources such as acetate and succinate, the PTS proteins are found in the phosphorylated form [6,11,17,18,39]. When glucose is transported, the phosphorylated IICB^Glc^ induces the expression of *ptsG* by binding and inactivating the expression of the global regulator of carbohydrate metabolism protein (Mlc) encoded by *mlc* [6,43]; when glucose or PTS sugars are exhausted from the medium, the P~IICB^Glc^/P~IIA^Glc^ ratio increases. P~IIA^Glc^ induces the activation of Cya (in the presence of an unknown Cya activation factor) [10,18], resulting in the conversion of ATP to cyclic AMP (cAMP). The cAMP concentration indicates glucose starvation in the culture medium [11]. cAMP forms a complex (as a coactivator) with CRP. The cAMP–CRP complex activates and increases the expression of genes repressed by the presence of glucose such as *crp,* the global regulator of carbohydrate metabolism *mlc*, and *ptsG* [11,17,18,23] (Figure 2A). Genes activated by the cAMP–CRP complex (catabolite repressed in the presence of glucose/absence of cAMP) have been estimated in a minimum of 378 in *E. coli* BL21(DE3), including genes for carbohydrate transport, genes coding for enzymes participating in reversible reactions of the central carbon metabolism, genes encoding rate controlling reactions in carbon metabolism, genes for side pathways linked to the central carbon metabolism, and some encoding RNAs [17,21]. Among them, more than 40 predicted transporting units under the control of CRP have been proposed, including 11 MFS transporters, 18 PTS transporters, and 9 ABC transporters [27]. 

The inducer exclusion mechanism controls the transport and phosphorylation of non-PTS sugars by the interaction of dephosphorylated EIIA^Glc^ (EIIA^Glc^) with lactose permease (LacY), the melibiose/H^+^/Na^+^/Li^+^ symporter (MelB), the galactose permease GalP, by interaction with the ATP binding subunit of the maltose ABC transporter (MalK), the D-galactose/methyl-galactoside (MglA), and the ribose ABC transporter (RbsA). Additionally, EIIA^Glc^ inhibits the glycerol kinase (GlpK) activity inhibiting the phosphorylation of intracellular glycerol [6,18,19] (Figure 1B). Inducer exclusion was initially associated with the inhibition of LacY, MelB, and GlpK by EIIA^Glc^ in cultures of *E. coli* in the presence of mixtures of glucose with these non-PTS carbohydrates. Nevertheless, the interaction of EIIA^Glc^ with the ATP binding subunit of other ABC transporters was determined to be stimulated by other PTS sugars associated with the phosphorylation state of EIIA^Glc,^ indicating that the inducer exclusion mechanism is a more general phenomenon. The interaction of EIIA^Glc^ with GalP was stimulated only by the presence of glucose but not by other PTS sugars [19]. 

The small regulatory RNA SgrS- RNA-binding protein Hfq (SgrS-Hfq complex) regulates the abundance of EIICB^Glc^ by repressing the translation of *ptsG* [11,44]. When *E. coli* K-12 was grown in high glucose concentration, expression of *sgrS* was lower, resulting in a higher PtsG concentration and higher glucose transportation. Nevertheless, this behavior was the opposite in *E. coli* BL21 with a higher *sgrS* transcription, resulting in a lower PtsG concentration and glucose uptake [44]. Additionally, the PtsG glucose transporter inhibitor StrT, localized in the cellular membrane, interacts with unphosphorylated PtsG and inhibits growth in minimal medium supplemented with glucose by inhibiting specifically PtsG, but not affects growth in the presence of other carbohydrates such as mannose, fructose, or trehalose [11]. The expression of the operon *sgrSsgrTsetA* is controlled by the DNA-binding transcriptional dual regulator SgrR. SgrT is translated under sugar-phosphate stress, and SetA is an efflux transporter capable of exporting several sugars such as glucose and lactose and sugar analogs; nevertheless, the physiological role of SetA is unclear [11,45] (Figure 2A).

#### 3.1.2. Regulatory Mechanisms of the PTS Proteins EI and Hpr

The phosphorylated state of EI and Hpr proteins determines several regulatory functions in *E. coli* [17,18,46]. Chemotaxis toward PTS sugars is controlled by dephosphorylated EI (EI). In the presence of PTS sugars or the absence of PEP, EI inhibits the activity of the chemotactic protein CheA promoting run swimming toward the PTS sugar [18].

Hpr was determined to interact allosterically with the glycolytic enzymes pyruvate kinase I (PykF) and glucosamine-6-phosphate deaminase (NagB), with the gluconeogenic enzyme 6-phosphofructokinase 2 (PfkB) and the adenylate kinase (Adk), an enzyme involved in the homeostatic control of AMP, ADP, and ATP [46]. Hpr regulates the activity of PfkB, a key enzyme in the glycolysis (fructose-6-P→fructose 1,6-diP), the D-allose, the D-mannose, the D-sorbitol degradation, and the galactitol degradation pathways, where this enzyme feeds into glycolysis [11]. This enzyme’s activity results in a decrement in the *K*_half_ for fructose-6-P, affecting both glycolysis and gluconeogenesis. Glucose-6-phosphate inhibits gluconeogenesis. Hpr activates PykF (PEP→PYR), resulting in the decrement of *K*_half_ for PEP, affecting glycolysis. NagB catalyzed the reaction of D-glucosamine 6-phosphate↔β-D-fructofuranose 6-phosphate in the *N*-acetylglucosamine degradation pathway. This enzyme is essential for the utilization of aminosugars. P~Hpr inhibits allosterically to Adk, an essential enzyme required for the biosynthesis of purine ribonucleotides, with a key role in controlling the rate of cell growth [11,46] (Figure 2B). 

Hpr but no EIIA^Glc^ controls the preferential consumption of glucose over other PTS sugars such as mannitol, sorbitol, galactitol, and fructose in *E. coli*. For mannitol control, Hpr accumulates during the transport of glucose and interacts with the mannitol transcriptional repressor MtlR, enhancing its repressor activity on the expression of the mannitol operon (*mltA* and *mltD,* encoding for the mannitol-specific PTS enzyme II and mannitol-1-phosphate 5-dehydrogenase, respectively), resulting in the inhibition of mannitol utilization [11,47] (Figure 2B). 

**Figure 2 microorganisms-11-01588-f002:**
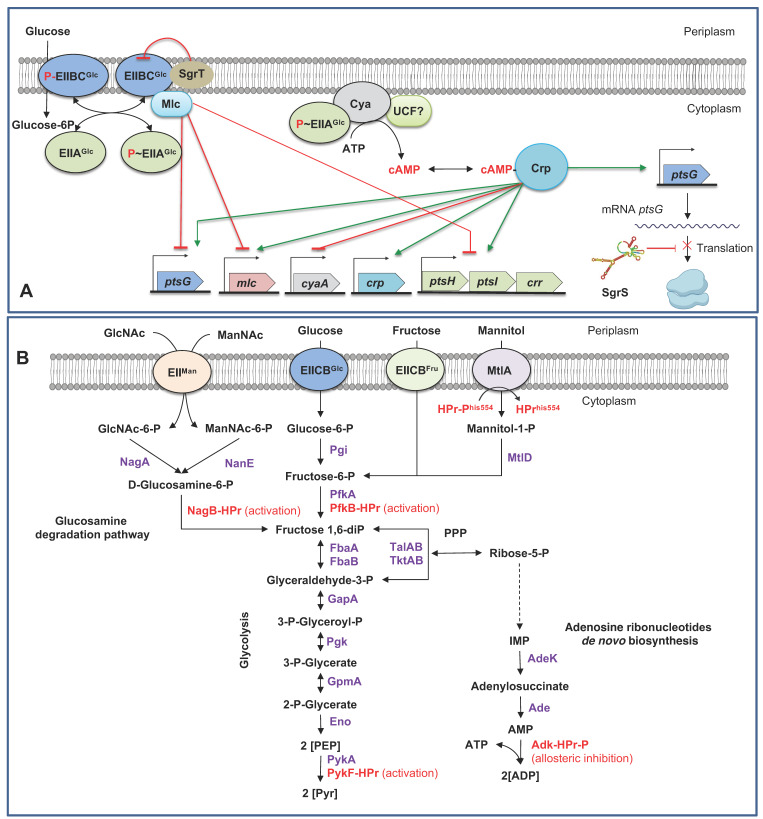
Regulatory mechanisms involved in PTS. (**A**). Regulatory, allosteric, and translational mechanisms of *ptsG*, *ptsHIcrr*, and PtsG. (**B**). Regulatory (activation and allosteric inhibition) by HPr in the transport of mannitol, in enzymes involved in glycolysis and gluconeogenesis, and in the cellular balance of the adenosine phosphates molecules. Enzymes: NagA, N-acetylglucosamine-6-phosphate deacetylase; NanE, predicted N-acetylmannosamine-6-phosphate 2-epimerase; NagB, glucosamine-6-phosphate deaminase; Pgi, glucose-6-phosphate isomerase; PfkA, 6-phosphofructokinase 1; PfkB, 6-phosphofructokinase 2, FbaA, fructose-bisphosphate aldolase class II, FbaB, fructose-bisphosphate aldolase class I, Pgk, phosphoglycerate kinase; GpmA, 2,3-bisphosphoglycerate-dependent phosphoglycerate mutase; Eno, enolase, PykA, pyruvate kinase II; PykF, pyruvate kinase I, MtlA, mannitol-specific PTS enzyme IICBA component; MtlD, mannitol-1-phosphate 5-dehydrogenase, AdeK, adenylosuccinate synthetase; Ade, amidophosphoribosyltransferase; Adk, adenylate kinase; TalAB, transaldolase A, transaldolase B; TktAB, transketolase I, transketolase II. Compounds: GlcNAc-6-P, *N*-acetyl-D-glucosamine 6-phosphate; GlcNAc, *N*-acetyl-D-glucosamine; IMP, inosine monophosphate; ManNAc-6-P, *N*-acetyl-D-mannosamine 6-phosphate; ManNAc, N-acetyl-D-mannosamine; PEP, phosphoenolpyruvate; PPP, pentose phosphate pathway; Pyr, Pyruvate. (**B**). Control of the expression of PTS proteins. SgrS, PtsG glucose transporter inhibitor; Mlc, DNA-binding transcriptional repressor; Cya, adenylate cyclase; UCF, unknown cytoplasmic factor; CRP, DNA-binding dual transcriptional regulator; cAMP, cyclic-AMP DNA-binding transcriptional dual regulator; SgrS, small regulatory RNA. The dotted lines indicated several enzymatic reactions. Green lines denote activation mechanisms. Red lines denote repression mechanisms. The secondary structure of SgrS was predicted in the ViennaRNA Package 2.0 [48]. Figure composed from references [6,11,17,18,19,21,22,23,43,44,46,49].

### 3.2. Primary and Secondary Active Transporters Can Transport Glucose in E. coli

#### 3.2.1. The ATP-Dependent Cassette (ABC) Transporters

This family of transporters comprises both import and export systems involved in the transport of diverse molecules from ions, sugars, and amino acids to large compounds such as drugs, antibiotics, or polypeptides across the membrane against a concentration gradient using energy derived from the hydrolysis of ATP to ADP [6,50,51]. In bacteria, two major ABC transporters are present, the prokaryotic type (PK-type) and the Eukaryotic-type (EK-type) [50]. The PK-type are importers, whereas the EK-type are exporters. In *E. coli* K-12, the ABC transporters are a large transporter family comprising ~5% of the total genome-encoding genes, comprising 71 up to 80 genes encoding this type of transporter [50,52] (Table 1). All transport proteins of the ABC type share four primary domains: two transmembrane domains, commonly referred to as the TMD transmembrane domain or the IMD integral membrane domain, and two cytosolic domains, called ABC or NBD (for its acronym, ATP-binding cassette domain or nucleotide-binding domain). The PK-type ABC importers consist of an additional extracellular element protein called the substrate-binding protein (SBP) or, specifically for Gram-negative bacteria, a periplasmic lipoprotein or periplasmic-binding protein (PBP). These components bind or recruit the substrate and transfer to the transporter [6,50,51]. 

In *E. coli* K12, there are 10 ABC transporters for sugar intake, including D-allose, D-ribose, L-arabinose, D-fructose, D-xylose, maltose, β-D-galactosides (galactose/glucose), and at least three unidentified sugars substrates [6,50] (Table 1). The maltose transporter ABC system (MalEFGK2, encoded by the contiguous and divergent *malEFG* and *malKlamBmalM* operons) is the best-known in *E. coli*. MalE is the PBP for maltose, MalF and MalG are the integral or transmembrane components (TMDs), and MalK2 (dimeric) is the ATP-binding component of the transporter [6,11]. Crystal structures, representing different states of the transport process, describe the maltose transport as an alternating transition between the solute uptake in the periplasm by MalE and releasing into periplasm: in a resting state, the MalF/MalG proteins are inward open, with the maltose binding pocket accessible from the cytoplasm. The ATP binding interface or domain (NBD) of dimeric MalK in a nucleotide-free or an ADP-binding environment is in an open conformation, and periplasmic MalE without sugar may be weakly attached or free to MalF/MalG. In the second stage, MalE captures maltose and bounds with high affinity to MalF/MalG. MalK releases ADP from two ATP molecules closing the dimer and induces the closing of the inward opened cavity of MalF/MalG, opening outward the cavity, which interacts with charged MalE, transferring maltose to MalF/MalG binding cavity. This transference induces a conformational change in the transmembrane components, triggering ATP hydrolysis in dimeric MalK. With ATP hydrolysis, dimeric MalK expands and exposes the MalF/MalG binding cavity inward open, releasing the substrate into the cytoplasm [6,11,53,54,55]. MalE has a higher affinity for oligomaltose than MalF/MalG, and, ligating to the complex, MalFGK2 stimulates the maximal ATPase activity of the complex [6,11,54,55]. Remarkably, the maltose ABC transporter can translocate glucose, and its expression is elevated under glucose-limited conditions in a chemostat [20].

The expression of the *malEFG* operon is induced by the maltotriose–MalT regulator complex and cAMP–CRP. MalK plays an additional role in controlling the activity of the maltose ABC transporter. When the substrate is absent and the system is resting (inward open cavity), MalT is sequestered by MalK, complexing them in the membrane and avoiding the interaction with maltotriose. MalT is released during the maltose uptake by MalEFGK2, binding with maltotriose and ATP, activating the transcription of the *malEFG* operon and *malK*. In the presence and transport of glucose and other PTS sugars, MalK is allosterically inhibited by EIIA^Glc^ by the inducer exclusion mechanism (Figure 1B) [6,11,56,57]. 

#### 3.2.2. The D-Galactose/Methyl-β-D-Galactoside ABC Transporter MglBAC

In *E. coli*, MglBAC is the high affinity (*K_m_* = 1 μM) D-galactose/methyl-D-galactoside uptake system composed by MglB, the periplasmic galactose/methyl-galactoside binding protein (and galactose/glucose chemoreceptor); MglC is the predicted integral membrane component, and MglA is the predicted ATP-binding component of the transporter complex [11,58,59]. MglBAC is one of the two transporters of galactose, together the galactose permease (GalP) of the galactose (*gal*) regulon, involved in the amphibolic utilization of D-galactose in the absence of glucose by the enzymes encoded by the *galETKM* operon in the absence of glucose [59,60].

The expression of the *gal* regulon is controlled by *galR* and *galS* encoding, the Gal repressor (GaIR) and the Gal isorepressor (GalS), respectively, and is activated by cAMP–CRP. The presence of D-galactose promotes the synthesis of GalR, and the active regulator exists in the complex D-galactose-GalR, which, in a dimeric structure, binds to the promoter regions of transporters *mglBAC* and *galP* in the presence of D-galactose and absence of D-glucose. However, in the absence of D-galactose, GalR represses the *galETKM* operon [11,59,60,61,62].

#### 3.2.3. The D-Galactose/H+ Symporter GalP

GalP is a member of the major facilitator superfamily (MFS), the largest known superfamily of secondary carriers in the biosphere. These secondary transporters facilitate the transfer of specific solutes down or against a concentration gradient. This membrane protein class catalyzes three types: uniporters, symporters, and antiporters. Nowadays, 89 families of MFS transporters are reported in TCDB [63]. In *E. coli*, there are three MFS transporters: the GalP, galactose/H^+^; AraE, arabinose/H^+^; and XylE, D-xylose/H^+^ symporters [51,64] (Table 1). GalP and MglABC are the primary transporters of D-galactose in *E. coli* K-12. GalP transports D-galactose by the sugar–proton symport mechanism [11,59,60,65]. GalP is induced by galactose or fucose and transports glucose, galactose, and fucose but not methyl *β*-galactosides [65]. 

## 4. Dynamics of Glucose Transport in *E. coli* under Sugar-Limiting Conditions

Despite the essential role of the PTS glucose system in glucose transport and phosphorylation in *E. coli*, as well as in controlling the preferential consumption of glucose over other non-PTS sugars, the cultivation of *E. coli* under nutritional stress conditions results in a differential expression and synthesis of other transport systems nor PTS Glc for glucose transport. Nutritional stress conditions under glucose-limited cultivation (1–300 μM, defined as a hunger condition) or under glucose starvation conditions (<0.1 μM) (growing under glucose-limited chemostat) result in lower specific growth rates (0.1–0.9 h^−1^) [33,34]. Under these scavenging conditions, *E. coli* activates the transcription and translation of alternative, high-affinity transporters for glucose such as several ABC transporters [33]. This process starts with the synthesis of the endogenous inducers galactose and maltotriose [33], which induces, respectively, the expression of the operon *mglBAC* (member of the *gal* regulon), the operons *malEFG (*maltose ABC transporter), the *malKlamBmalM* operon encoding for the maltose ABC transporter ATP binding subunit (MalK), the maltose outer membrane channel/phage lambda receptor protein (LamB), and the maltose regulon periplasmic protein (MalM), both part of the *mal* regulon [11,33]. Additionally, the glucose limitation condition results in elevated levels of cAMP compared to the concentration when growing in high glucose concentration, activating the expression of the above operons [33,34,66] (Figure 3). 

When growing at limiting micromolar glucose concentrations (hunger response), outer membrane porins in *E. coli* (mainly OmpF/OmpC) can permeate glucose. However, the affinity of LamB for carbohydrates selects this OMP as the primary way to introduce extracellular glucose to the periplasm [33,34]. Induction of the *malKlamBmalM* operon by maltotriose-MalT (DNA-binding transcriptional activator MalT-maltotriose-ATP) induced under glucose limitation increases expression of *lamB*. This condition suggests an increased concentration of LamB in the outer membrane and an increased concentration of periplasmic glucose, which is then transported into the cytoplasm by MglBAC and MalEFG transporters. The availability of the inducer D-galactose inactivates both the GalR repressor and the DNA-binding transcriptional dual regulator GalS (Figure 3), allowing the expression of *mglBAC* by cAMP–CRP [11,60,70], and the *malEFG* operon is induced by the presence of maltotriose-MalT and the cAMP–CRP complex [11]. Overexpression of the highly sensitive glucose transportation system comprising the *malKlamBmalM* and *mglBAC* operons showed a higher expression level during the hunger response (LamB 60X, MglBAC 20X, and OmpF 20X). However, the expression level of these operons was lower in starvation conditions (LamB 5X, MglBAC 1X, and OmpF 7X) compared to the expression level growing in glucose-rich conditions [34,67,68]. Increased expression and translation of *ompF* were observed under glucose limitation at D = 0.3 h^−1^ in glucose- or nitrogen-limited chemostat cultures [69].

Inactivation of PTS glucose in *E. coli* for the selection of mutants avoiding PEP consumption for aromatic compounds production purposes [71,72] imposes a severe nutritional stress condition when PTS^−^ mutants are grown in glucose as the sole carbon source, resulting in a severe decrement the specific growth rate of PTS^−^ mutants (Table 2).

The use of adaptive laboratory evolution (ALE) experiments for the selection of fast-growing derivatives in glucose from PTS^−^ mutants resulted in derivative mutants that increased the specific growth compared to the parental PTS^−^ mutants, developing several mutations, resulting in the selection of alternative glucose transporter systems to PTS glucose [71,74,75,76,77]. The characterization of evolved mutants derived from *E. coli* JM101, W3110, and MG1655 selected GalP as the primary glucose transporter for the phosphorylation of incoming glucose by Glk (glucokinase) from ATP [74,75,78,79]. The selection of GalP as the glucose transporter in evolved PTS^−^ derivatives resulted in the inactivation of the transcriptional repressor of the *gal* regulon GalR by the complete or partial deletion of *galR* or the selection of mutations resulting in the inactivation of the function of the repressor [5] (Figure 4). The selection of GalP for glucose transport in PTS^−^ evolved mutants results during the ALE experiment. In the ALE experiment of the PTS derivative from JM101, the evolving population grew exponentially after 75 h of cultivation. The transcriptional analysis of the evolved derivative mutant selected after 120 h of cultivation showed elevated transcription values for *galP* (13.1X), but a decrement in the transcript of *galR* (1.2) and *galS* (3.2X), suggesting the derepressing of *galP* and the synthesis of the transporter GalP [73,74]. The appearance of a mutated version of *galR* (deletion of the 72-bp region) was reported between 48–72 h of cultivation during an ALE experiment [80]. The proteomic analysis of the PTS mutant from JM101 and the evolved derivative PB12 mutant showed an increased concentration of LamB, ManX, and MglB in the parental PTS^−^ mutant (μ = 0.13 h^−1^) compared to the observed protein concentration in the evolved derivative PB12 (μ = 0.44 h^−1^). The concentration of LamB and MglB decreased in the evolved mutant, suggesting that MglBAC was selected as the glucose transporter in the Δ*ptsHIcrr* mutant and during the first 50–75 h of the ALE experiment [77], which was replaced by GalP during the evolution experiment [73,75,77,80] (Figure 4).

The resultant, evolved, fast-growing derivative mutants from the ALE experiments recovered their specific growth rate to μ values ranging from 0.2 to 0.92 h^−1^ [5,74,75,78,79,80]. Nevertheless, the μ values in these fast-growing PTS^−^ mutants were consistently lower than those observed μ in the parental PTS^+^ strains. The selection of alternative glucose transporters in the absence of PTS glucose involved the dependence of ATP to move one molecule of glucose across the inner membrane by the ABC transporters such as MglBAC, and the further phosphorylation of incoming glucose by Glk also from ATP, to yield glucose-6-P (total ATP cost = 2). The cotransportation of one molecule of glucose and one H^+^ by GalP and the phosphorylation of imported glucose by Glk from ATP had an additional energetic cost because ATP synthase needed to hydrolyze ATP to maintain the proton gradient, resulting in a total ATP cost ≥ 1.25 [28,74,75]. In these mutants, the dependence of ATP for glucose transport and phosphorylation entailed a reduction in ATP and an increment of AMP levels with a decrement in energy availability, resulting in reduced growth rates [75]. 

## 5. Transport Engineering or Improving Sugar Uptake Capabilities for Metabolic Engineering Purposes

Sugar transport is essential for heterotrophic bacteria such as *E. coli* to fuel carbon for biosynthetic metabolism and growth. Glucose is the primary substrate used as a carbon source for the industrial cultivation of *E. coli* for production purposes. Nevertheless, the use of PTS glucose as the preferential transport system comprises the availability of one molecule of PEP as a metabolic precursor (e.g., for aromatic production purposes) and imposes CCR or inducer exclusion mechanisms, avoiding the simultaneous consumption and co-fermentation of sugar mixtures. The inactivation of PTS glucose by *ptsHIrr* operon KO, deletion of specific genes involved in phosphocarrier from PEP (*ptsH*) [81], and inactivation of *ptsG* have been reported as the main strategies for inactivation of PTS in engineered strains for shikimic acid (SA) production, a valuable aromatic compound used as the precursor for the chemical synthesis of the antiviral oseltamivir phosphate [73,74,75]. Leading strategies to overcome the decrement in growth in resulting *E. coli* PTS^−^ mutants and to restore their growth capability in glucose were the cloning of the glucose-facilitated diffusion protein Glf with Glk from *Zymomonas mobilis* or the selection of fast-growing mutants in glucose, which selected GalP as the primary glucose transporter during ALE experiments.

Further genetic modifications in these PTS^−^ mutants resulted in overproducing strains for SA [71,72] (Table 3). Remarkably, resulting PTS^−^ mutants abolishing the CCR or inducer exclusion mechanisms can simultaneously use glucose and xylose mixtures as carbon sources for production metabolite purposes such as succinate and cis, cis-muconic acid, demonstrating that product yield and productivity were significantly improved in these PTS^−^ mutants (Table 3). The overexpression of GalP in the Δ*galP* derivative strain of *E. coli* KJ122 showed a negative regulatory control mechanism of the expression of catabolic genes of other sugars such as xylose and arabinose, suggesting that the selection of GalP as the leading glucose transporter in the PTS^−^ derivative was an efficient strategy [28,82].

Engineering of alternative transport systems includes the permeabilization of the Sec translocon subunit SecY, which, in *E. coli*, is the core component of the heterotrimeric SecYEG preprotein translocase responsible for protein translocation [11,83]. The permeabilization of SecY by the deletion of the plug domain and mutations in the pore ring allows for the passage of glucose, xylose, mannose, arabinose, and lactose. The engineered SecY *E. coli* derivative showed rapid growth, avoiding transport saturation, sugar-specific saturation, and CCR mechanisms, resulting in an attractive strategy for sugar transport engineering for developing engineered strains [83].

**Table 3 microorganisms-11-01588-t003:** Relevant transport engineering strategies employed in PTS^−^ mutants of *E. coli* for glucose transport or abolishing of CCR mechanisms.

Derivative Mutant and (Parental Strain)	Mutation	Alternative Glucose Transport System	Resultant Phenotype or Metabolic Engineering Applications	References
SP1.1 pSC6.090B (RB791 derivative)	*ptsHIcrr* KO	Glf from *Zymomonas mobilis.*	Cloning of Glk from *Z. mobilis* improved glucose phosphorylation. Genetic background for SA overproduction: 87 g/L of SA in 36% mol SA/mol Glc yield in the final derivative.	[72]
PB12 (JM101)	*ptsHIcrr* KO	GalP was selected as the leading glucose transporter during an ALE experiment. Selection of MglBAC during the early ALE experiment as primary glucose transporter.	Abolition of CCR mechanism. Increased glycolytic fluxes of 93.1% compared to the parental strain. Higher Glk activity. Genetic background for SA overproduction: SA titer of 42–60 g/L in 42% mol SA/mol Glc yield.	[71,74,77,84]
YL104H (MG1655)	*ptsH* KO	Not described	Abolition of CCR mechanism. PTS glucose mutant for succinate production in cultures with glucose: xylose mixtures under anaerobic conditions. Succinic acid production of 511.11 mM and 1.01 g/L/h.	[81]
SB2/pPckA (MG1655)	*pts*H KO	Glf from *Z. mobilis*	Resultant PtsG^−^ Glf*^Z. mobilis^* mutant increased succinate yield by 489. 65 X compared to the parental strain.	[85]
STG8 (*E. coli* W KCTC1039)	Δ*ptsG,* Δ*malE*, Δ*mglB*, Δ*Gal*P	Upregulation of remaining functional PTS sugar systems and ABC transporters.	Delayed glucose consumption, extended lag phase, low or no acetate production. Upregulation of PTS systems: Trehalose, glutitol/sorbitol, Mannose/fructose/ sorbose/D-GalNAc, UDP-GlcNAc, N-acetylmuramic acid. Upregulation of ABC systems: Arabinose, glycerol- 3-P, ribose, xylose, gluconate, hexuronate. Derivative STG8 strain increased the yield for EGFP to 132%, GABA titer to 130% with increased specific yield of 176%, increased lycopene yield of 90%	[86]
CFT5 (*E. coli* ATCC31882)	*ptsG* KO by replacing with galactose permease/glucokinase	Transport of glucose by GalP.	Abolition of the CCR mechanism, simultaneous use of glucose and xylose as carbon sources, independence of glycolysis and PPP from TCA. The heterologous Dahms pathway channeled xylose into TCA, glucose transported by GalP channeled to cis, cis-muconic acid production.	[87]
Several derivatives from W3110	WG, *ptsG* KO; WGM, *manX* KO; WGMC, *mglBAC* KO; WHIC, *ptsHIcrr* and *mglBAC* KO	Glucose transport by alternative systems: WG: IICD^Man^, MglBAC; WGM: MglBAC; WGMX, Unknown; WHIC, Unknown. Differential upregulation of other PTS:sugar systems, non-PTS sugar transporters, and catabolic proteins for several sugars.	Mutants with reduced μ values compared to W3110, lower acetate production. Increased transcription in genes of alternative sugar transport and metabolism, energy generation, and amino acid biosynthesis in WG derivative compared to W3110: Upregulated PTS systems: D-GalNAc, fructose, galactitol, mannose, mannitol, glucitol/sorbitol, UDP-GlcNAc, trehalose. Upregulated non-PTS transporters: maltose, ribose, galactose/glucose, arabinose, inositol, 2-D-3-deoxygalactose, fuculose, 5-keto-4-deoxy-D-glucarate and 2-keto-3-deoxy-D-glucarate, tagatose, maltose and maltodextrin, ManNAc, 2-methylisocitrate, glucuronate altronate, mannonate, 5-dehydro-4-deoxy-D-glucuronate.	[88]

CCM, Central carbon metabolism; EGFP, Enhanced green fluorescent protein; GalNAc, *N*-acetyl-D-galactosamine; GlcNAc, N-acetyl-D-glucosamine; ManNAc, N-acetyl-D-mannosamine; TCA, tricarboxylic acids cycle; PPP, pentose phosphate pathway; SA, shikimic acid.

## 6. Conclusions and Perspectives

Glucose is the main carbohydrate used as the carbon and energy source for the cultivation of *E. coli* in laboratories and the industrial cultivation of diverse derivatives in large-scale biotechnological processes. *E. coli* diffuses glucose from the extracellular medium into periplasm via the OMPs OmpF, OmpC, and LamB. Periplasmic glucose is internalized into the periplasm by active transport systems, where the PTS glucose system is used to transport glucose and phosphorylate it from PEP to yield intracellular glucose-6-P. Growing in carbon limitation conditions or in the absence of PTS (Δ*ptsG* or Δ*ptsHIcrr*), *E. coli* transports glucose by other backup transporters such as the ABC systems dependent on ATP such as MglBAC or by the ion-gradient driven mechanisms such as GalP, with the further phosphorylation of cytoplasmic glucose from ATP by Glk. Nevertheless, using these backup glucose transporter systems results in lower specific growth rates due to the consumption of ATP.

PTS glucose imposes the strict control mechanisms of CCR and inducer exclusion, controlling the expression of genes involved in the transport and catabolism of secondary non-PTS sugars, including the activity of other PTS systems for sugars other than glucose. These traits of PTS glucose represent a significant limitation when *E. coli* is cultivated in carbon sources mixtures such as those resulting from the hydrolysis of cellulolytic biomass, containing both glucose and xylose, or for overproduction purposes of metabolites derived from the PEP. Diverse strategies to improve the overall fermentative capabilities to produce natural or heterologous compounds in diverse engineered derivatives of *E. coli* have focused on sugar transport engineering as a fundamental strategy for metabolic engineering purposes.

The inactivation of PTS glucose (Δ*ptsG* or Δ*ptsHIcrr*) as the primary glucose transport system for the cultivation of *E. coli* results in the abolishing of CCR and inducer exclusion mechanisms, allowing for the simultaneous transport and catabolism of sugar mixtures. The replacement of PTS glucose by other glucose transporters in PTS^−^ derivatives of *E. coli* by strategies such as cloning the native GalP; heterologous systems for glucose transport and phosphorylation, such as Glf and Glk from *Z mobilis*; and the selection of fast-growing mutants merged from AEL experiments have restored the transport of glucose for diverse metabolite production purposes efficiently. The complementation of glucose transport and catabolism, together improving the transport of other sugars such as xylose by native XylE in PTS^−^ mutants of *E. coli* and the introduction of heterologous xylose catabolic pathways (such as the Dahms pathway), has resulted in the efficient differential coutilization of both sugars from lignocellulosic hydrolysates [87]. This derivative channeled carbon from glucose for target metabolite production and xylose for carbon supply for growth provides the basis for efficient consumption and targeted sugar catabolism in sugar mixture utilization. 

OMPs comprise ~50% of overall proteins anchored in the outer membrane in Gram-negative bacteria, but *E. coli* was demonstrated to be able to grow in the absence of these proteins, suggesting the possibility of modulating the proportion of OMPs such as OmpF and OmpC to modify the cell permeability under specific growth conditions. The engineering of OMPs focused on improving channel properties such as geometry and diameter for substrate specificity and solute diffusion purposes represents an attractive but unexplored strategy to improve sugar and another substrate uptake. Finally, engineering alternative transport systems such as SecY is an attractive strategy to improve substrate specificity to extracellular solutes and bypass the metabolic and regulatory mechanisms associated with PTS.

## Figures and Tables

**Figure 1 microorganisms-11-01588-f001:**
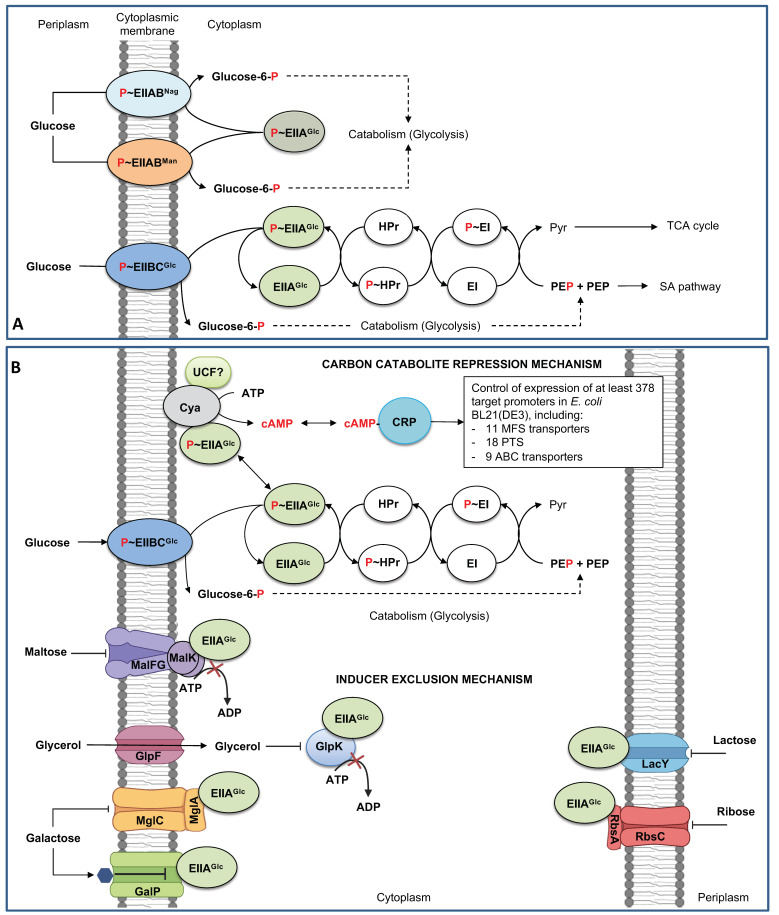
The PTS glucose in *Escherichia coli*, the carbon catabolite repression, and inducer exclusion mechanisms. (**A**). Components and function of PTS glucose. Alternative glucose transport and phosphorylation by EIIAB^Man^ and EIICBA^Nag^. (**B**). Carbon catabolite repression and inducer exclusion mechanisms. cAMP, cyclic-AMP DNA-binding transcriptional dual regulator; CRP, DNA-binding transcriptional dual regulator; Cya, adenylate cyclase; GalP, galactose permease; GlpF, glycerol facilitator; GlpK, glycerol kinase; LacY, lactose permease, MalFG, maltose ABC transporter membrane subunits F and G; MalK (dimeric), maltose ABC transporter ATP binding subunit; PEP, phosphoenolpyruvate; Pyr, pyruvate, RbsA, ribose ABC transporter ATP binding subunit; RbsC, ribose ABC transporter membrane subunit; TCA, the tricarboxylic acid cycle. The hexagon in GalP indicated a galactose; red-labeled P indicates a phosphate group in the phosphotransference mechanism. P~ indicates phosphorylated forms of PTS proteins. SA, shikimic acid. The dotted lines indicated several enzymatic reactions. **⊥** shows interrupted mechanisms or reactions. Figure composed from [6,11,17,18,19,20,21,22,23].

**Figure 3 microorganisms-11-01588-f003:**
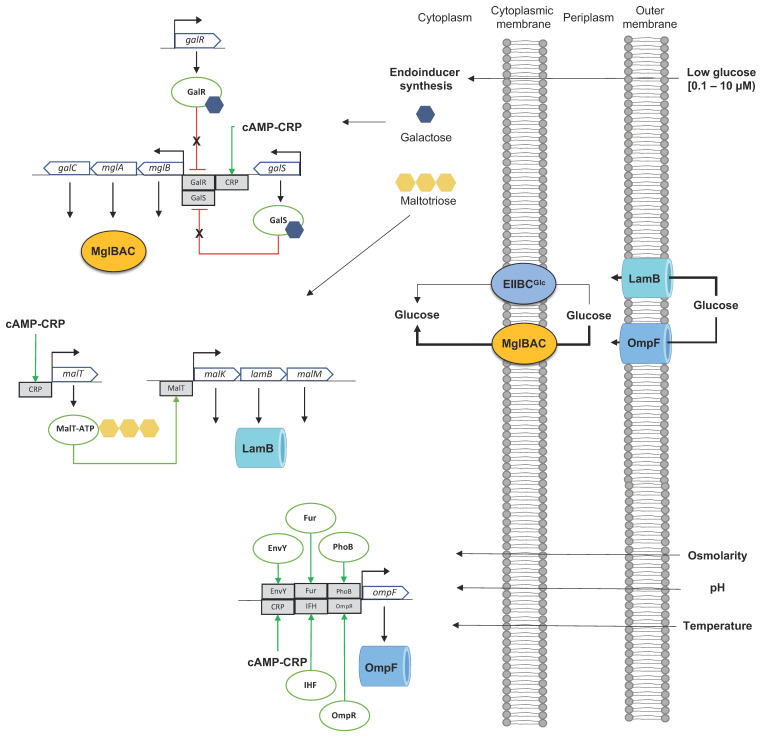
Induction of high-affinity glucose transporter growing in glucose-limiting conditions. Growing *E. coli* under glucose-limiting chemostat conditions induces the expression of the high-affinity glucose transporters MglBAC and OMP LamB by the coordinate action of cAMP–CRP and the autoinducers galactose or maltotriose. Induction of the *mglBAC* operon: autoinducer galactose (blue hexagon) binds to negative transcriptional repressors of the *gal* regulon GalR and GalS, inactivating them. cAMP–CRP binds to the DNA-binding transcriptional region of the *mglBAC* operon (gray rectangle), inducing its transcription and translation. Induction of the *malKlamBmalM* operon: autoinducer maltotriose (yellow triple hexagons) binds to the DNA-binding transcriptional activator MalT, inhibiting the repression of the operon, resulting in a higher transcription and the synthesis of encoding proteins, including LamB. Bold lines in transporters indicate increased glucose transport resulting from higher protein concentration. LamB increases glucose permeability to the periplasm, and MglBA acts as a high-affinity glucose transporter from the periplasm to the cytoplasm. EnvY, DNA-binding transcriptional activator EnvY; Fur, DNA-binding transcriptional dual regulator Fur; IHF, integration host factor; OmR, OmpR dimer; PhoB, DNA-binding transcriptional dual regulator PhoB. Red lines show repression mechanisms. Green arrows induction mechanisms. Gray rectangles show specific DNA-binding transcriptional regions. Figure composed from references [11,15,33,34,67,68,69].

**Figure 4 microorganisms-11-01588-f004:**
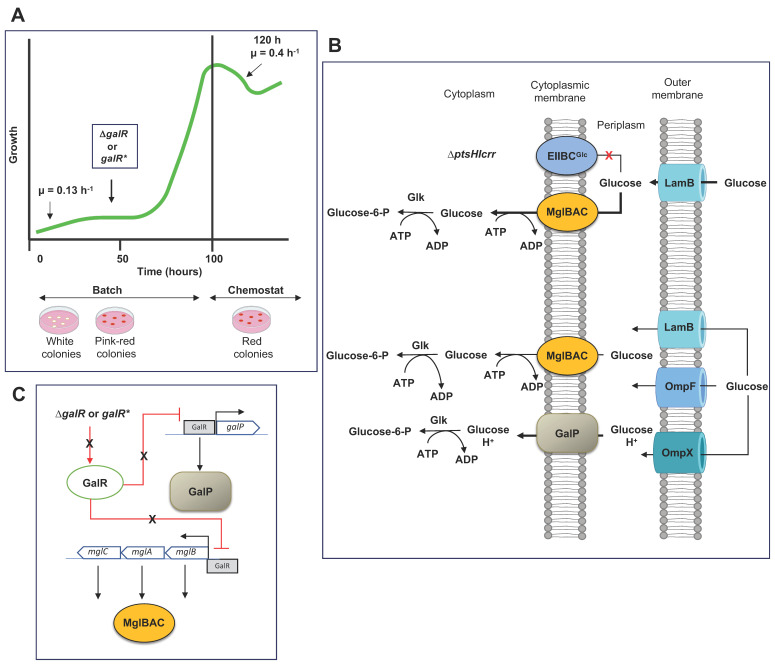
Selection of alternative glucose transporters during adaptive laboratory (ALE) experiments of a Δ*ptsHIcrr mu*tant (PTS^−^) of *E. coli* JM101. (**A**). ALE experiment with two-stage batch-chemostat stages in M9 minimal medium supplemented with glucose. The bold green line shows the overall growth profile starting with a μ = 0.1–0.13 h^−1^. PTS mutants in the early stage of the ALE experiment showed a white-color phenotype in MacConkey agar supplemented with glucose selecting LamB to diffuse extracellular glucose from the extracellular medium into the periplasm and MglBAC to transport glucose from the periplasm into the cytoplasm. (**B**). Analysis of several intermediate mutants indicated that in the absence of PTS (Δ*ptsHIcrr*) mutants in the early stages of the ALE experiment selected, MglBAC for glucose transport (**upper** section). After 100 h of cultivation, the ALE experiment switched on a chemostat stage, isolating red colonies with increased μ values. Fast-growing mutants showed a μ = 0.4 h^−1^, and further analysis showed that mutants selected GalP as the primary glucose transporter (**bottom** section). Bold arrows show a higher glucose transport. (**C**) shows the regulatory mechanisms resulting in the selection of MglBAC and GalP as alternative glucose transporters without the activity of PTS glucose. Proposed induction and synthesis mechanisms for LamB are illustrated in Figure 3. *galR**, mutated *galR*. Glk, glucokinase. OmpF, outer membrane porin; F OmpX, outer membrane porin X or OmpP. Red lines show repression mechanisms. Gray rectangles show specific DNA-binding transcriptional regions. This figure was composed of references [11,20,73,74,77,80].

**Table 1 microorganisms-11-01588-t001:** Carbohydrate transport systems in *Escherichia coli* K12 substr. MG1655.

Gene(s)	Transporter Family	Transported Sugar	PROTEINS	Cellular Location
*alsBAC*	ABC	D-allose	D-allose ABC transporter membrane	P, IM, C
*araFGH*	ABC	L-Arabinose	Arabinose ABC transporter	P, IM, C
*malEFG-malK*	ABC	Maltose/maltodextrine	Maltose ABC transporter	P, IM, C
*malK*	ABC	Maltose/maltotetraose/ maltotriose	Maltose ABC transporter ATP binding subunit	IM
*mglBAC*	ABC	D-galactose/methyl-galactoside	D-galactose/methyl-galactoside ABC transporter	P, IM, C
*rbsACB*	ABC	Ribose/D-xylose	Ribose ABC transporter	P, IM
*upgBAEC*	ABC	*sn*-Glycerol 3-phosphate	*sn*-Glycerol 3-phosphate ABC transporter	P, IM, C
*xylFHG*	ABC	D-Xylose	Xylose ABC transporter	P, IM, C
*yphFED*	ABC	Sugar	Putative ABC transporter	P, IM
*ytfQRT-yjfF*	ABC	β-D-Galactofuranose α-D-Galactofuranose	Galactofuranose ABC transporter	P, IM
*araE*	MFS (SP)	Arabinose	Arabinose:H^+^ symporter	IM
*dgoT*	MFS (ACS)	D-Galactonate	D-Galactonate:H^+^ symporter	IM
*fucP*	MFS (FHS)	L-Fucose/D-arabinose/ L-galactose	L-fucose:H^+^ symporter	IM
*galP*	MFS (SP)	D-Galactose	Galactose:H^+^ symporter	IM
*garP*	MFS (ACS)	Galactarate/D-glucarate	Galactarate/D-glucarate transporter	IM
*glpT*	MFS (OPA)	Glycerol-3-phosphate	sn-glycerol 3-phophate:phosphate antiporter	IM
*gudP*	MFS (ACS)	Galactarate/D-glucarate	Galactarate/D-glucarate transporter	IM
*lacY*	MFS (OHS)	Lactose/melibiose	Lactose/melibiose:H^+^ symporter	IM
*lgoT*	MFS (ACS)	L-Galactonate	L-Galactonate:H^+^ symporter	IM
*setA*	MFS (SET)	Lactose	Sugar exporter SetA	IM
*setB*	MFS (SET)	Lactose	Sugar exporter SetB	IM
*setC*	MFS (SET)	Arabinose-like	Putative arabinose exporter	IM
*uhpC*	MFS (OPA)	Sugar phosphate	Inner membrane protein sensing glucose-6-phosphate	IM
*uhpT*	MFS (OPA)	Hexose-6-phosphate	Hexose-6-phosphate:phosphate antiporter	IM
*xylE*	MFS (SP)	Xylose	D-xylose:H^+^ symporter	IM
*ydeA*	MFS (DHA1)	Arabinose	L-arabinose exporter	
*agaBCD*	PTS	Galactosamine	Galactosamine specific PTS system EIIBCD	IM, C
*agaV*	PTS	n-acetyl-D-galactosamine (galactose)	N-acetyl-D-galactosamine specific PTS system IIB	C
*ascF*	PTS	β-Glucoside (arbutin/cellobiose/salicin)	β-Glucoside specific PTS enzyme IIBC	IM
*bglF*	PTS	β-Glucoside (metil-β-D-glucoside, arbutine, salicin, β-D-glucose)	β-Glucoside specific PTS enzyme II/BglG kinase/BglG phosphatase	IM
*chbAC*	PTS	β-D-Cellobiose/chitobiose (starch, sucrose)	N, N’-diacetyl chitobiose-specific PTS enzyme IIAC	C
*chbB*	PTS	β-D-Cellobiose/chitobiose (starch, sucrose)	N, N’-diacetyl chitobiose-specific PTS enzyme IIB	IM
*cmtA*	PTS	Mannitol (fructose and mannose)	Mannitol-specific PTS enzyme IICB	IM
*cmtB*	PTS	Mannitol (fructose and mannose)	Mannitol-specific PTS enzyme IIA	C
*fruA*	PTS	Fructose and mannose	Fructose-specific PTS multi-phosphoryl transfer protein FruA PTS system EIIBC	IM
*frvA*	PTS	Fructose-like	Putative PTS enzyme IIA	C
*frvB*	PTS	Fructose-like	Putative PTS enzyme IIBC	IM
*frwB—frwD*	PTS	Fructose-like	Fructose-like PTS system EIIB	C
*frwC*	PTS	Fructose-like	Fructose-like PTS system EIIC	IM
*fryC*	PTS	Fructose-like	Fructose-like PTS system EIIC	IM
*fryB*	PTS	Fructose-like	Fructose-like PTS system EIIB	C
*gatA*	PTS	Galactitol	Galactitol-specific PTS system EIIA	C
*gatB*	PTS	Galactitol	Galactitol-specific PTS system EIIB	C
*glvBC*	PTS	α-Glucoside	Alpha-glucoside PTS system EIICB	IM
*malX*	PTS	Maltose/glucose	PTS enzyme IIBC component MalX	IM
*manYZ*	PTS	Mannose	Mannose-specific PTS system EIICD	IM
*manX*	PTS	Mannose	Mannose-specific PTS system EIIAB	IM, C
*mngA*	PTS	2-O-α-mannosyl-D-glycerate	2-O-α-mannosyl-D-glycerate specific PTS enzyme IIABC	IM
*mtlA*	PTS	Mannitol (fructose, mannose)	Mannitol-specific PTS enzyme IICBA	IM
*nagE*	PTS	n-Acetylglucosamine	N-acetylglucosamine-specific PTS enzyme II	IM
*ptsG*	PTS	Glucose	Glucose-specific PTS enzyme IIBC component	IM
*ptsHIcrr*	PTS	Glucose	*ptsH*, phosphor carrier protein HPr *ptsI*, PTS enzyme I *crr*, Enzyme IIA^Glc^	C
*sgcA*	PTS	Galactitol-like	Galactitol-specific PTS system EIIA	C
*sgcB*	PTS	Galactitol-like	Galactitol-specific PTS system EIIB	C
*sgcC*	PTS	Galactitol-like	Galactitol-specific PTS system EIIC	IM
*srlA*	PTS	Glucitol/Sorbitol	Sorbitol specific PTS system IIC_2_	IM
*srlB*	PTS	Glucitol/Sorbitol	Sorbitol specific PTS system EIIA	C
*srlE*	PTS	Glucitol/Sorbitol	Sorbitol specific PTS system IIBC_1_	IM
*treB*	PTS	Trehalose	Trehalose-specific PTS enzyme IIBC	IM
*ulaABC*	PTS	Ascorbate	L-ascorbate specific PTS system EIICBA	IM, C
*bglH*	OT (C/P)	β-Glycosides	Carbohydrate-specific outer membrane porin, cryptic	OM
*glpF*	OT (MIP)	Glycerol	Glycerol facilitator	IM
*lamB*	OT (C/P)	Maltose	Maltose outer membrane channel/phage lambda receptor protein	OM
*melB*	OT (EDP)	Melibiose	Melibiose:H^+^/Na^+^/Li^+^ symporter	IM
*ompF*	OT (C/P)	Sugar	Outer membrane porin F	OM
*ompC*	OT (C/P)	Sugar	Outer membrane porin C	OM

Transport mechanisms: ABC, ABC transporter system; MFS, Major facilitator superfamily (SP, Sugar porter family; OHS, Oligosaccharide symporter family; FHS, Fucose symporter family; SET, sugar efflux transporter; DHA1, The drug H^+^Antiporter-1; OPA, Organophosphate.Pi antiporter; ACS, Anion/cation symporter); PTS, PTS transporter system; OT, Other transporters (MIP, The major intrinsic protein (aquaporin); C/P, Channels and pores; EPD, Electrochemical potential-driven transporters). Cellular location: OM, Outer membrane; P, Periplasm; IM, Inner membrane; C, Cytoplasm. Table elaborated from data available in the EcoCyc database (https://ecocyc.org, accessed on 1 May 2023) [11].

**Table 2 microorganisms-11-01588-t002:** Nutritional stress conditions imposed in several *E. coli* strains resulting from the inactivation of PTS.

Parental Strain	PTS Mutation	Growth and Relevant Changes in the Expression of Several Genes Involved in Transport Respect the Parental Strain		References
MG1655	Δ*ptsG*	Aerobic conditions	Anaerobic conditions	[20]
		Decrement in μ of 73%. Increased expression of *galS* and down-regulation of *galP* (0.2 X) and *manX* (0.5 X). Overexpression of the *mgl* operon in 10 X. Downregulation of *cyaA* and increased levels of cAMP: 552.5 X.	Decrement in μ of 70.2%. Increased expression of *galS* and downregulation of *galP.* Increased expression of *malE* (48 X). Overexpression of the *mgl* operon in 48 X. Down-regulation of *cyaA* with increased levels of cAMP: 390.9 X.	
JM101	Δ*ptsHIcrr*	Reduction in μ~85% to 57%.		[73,74]
		Overexpression of *mglB* 13.4 X and *lamB* 17.6 X.		
		Overexpression of some genes of the *gal* regulon: *galP* 12.4 X, *galR* 3.2X, *galS* 4.9X.		
MG1655	*ptsHIcrr* KO	Reduction in μ~79%.		[75]

## Data Availability

Data sharing not applicable.

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
