# Peer review of "Glucose Transport in Escherichia coli: From Basics to Transport Engineering"

_microorganisms, 2023, doi:10.3390/microorganisms11061588_

Round 1

Reviewer 1 Report

This is a systematic review on glucose transport in E. coli, which covers the glucose transport process, the structure and mechanism of the PTS system,  the glucose trannsporters, the dynamics of glucose transport under sugar-limiting conditions, and engineering of sugar transport for metabolic engineering purposes.  Below are some comments:

1. Glucose transportation in particular via the PTS system is a classical process, so the authors are suggested to mention whether there are previous reviews on this topic. If so, it would be necessary to emphasize the difference of the present review from the previous ones.  

2. Line 57: "do not reports" should be "do not report".

3. "E. coli" should be italic.

Author Response

This is a systematic review on glucose transport in E. coli, which covers the glucose transport process, the structure and mechanism of the PTS system,  the glucose trannsporters, the dynamics of glucose transport under sugar-limiting conditions, and engineering of sugar transport for metabolic engineering purposes.  Below are some comments:

Reply: Dear reviewer, thanks for the suggestions and comments on the first version of our contribution. Please find the point-by-point reply for each comment. All the changes in the new version of the manuscript are highlighted in the red text in the corresponding lines.

  1. Glucose transportation in particular via the PTS system is a classical process, so the authors are suggested to mention whether there are previous reviews on this topic. If so, it would be necessary to emphasize the difference of the present review from the previous ones.

Reply: Dear Reviewer, thanks for the concern. Regarding the scope of this contribution, we searched the topics sugar+glucose+transport+E. coli in the journal Microorganisms, and we found no match covering these topics. Based on this result, we proposed the current review for publication in the journal.  Previous reviews on PTS and sugar transport in E. coli were published, including one recent contribution of our authorship: DOI 10.1007/s00253-019-10335-x (New insights into transport capability of sugars and its impact on growth from novel mutants of Escherichia coli, 2020), https://doi.org/10.1007/978-3-030-18768-2_8 (Carbohydrate Transport by Group Translocation: The Bacterial Phosphoenolpyruvate: Sugar Phosphotransferase System, 2019), and https://doi.org/10.1007/s00424-020-02379-0 (Transporters of glucose and other carbohydrates in bacteria, 2020).

Because of the relevance of glucose as the principal carbon source used for laboratory and industrial fermentations with diverse strains of Escherichia coli, our submitted contribution to Microorganisms-MDPI covers specifically the structure and mechanisms of the glucose transporter systems in Escherichia coli, including the regulatory circuits involved in their expression under specific growing conditions. As PTS:glucose plays a crucial role in the physiology and adaptation of E. coli when growing in sugar carbon mixtures, particularly with the catabolite control repression, inducer exclusion mechanisms, and the specific regulatory role of the PTS components EI and Hpr, we covered this topic in greater depth, including the mechanisms for controlling the expression of this system, which are not reviewed in the previous reviews mentioned above.

We also described the relevant structural, physiological, and regulatory characteristics of the ABC and MFS systems for glucose transport. Finally, we described how E. coli contends with their transport and physiological capabilities when the PTS:glucose system is disrupted, and the cell is exposed to a nutrient scavenging condition, selecting alternative glucose transporters. In the last section of the contribution, we described selected examples of the successful impact of transport engineering in diverse E. coli derivatives for the overproduction of diverse metabolites and the simultaneous consumption of several carbon sources for overproduction purposes. As a critical contribution to our understanding, we updated a complete list of all the sugar transporter systems described in E. coli K12 substrain MG1655, shown in Table 1. The above description of the scope of this contribution was not covered in the previously mentioned published reviews.

  1. Line 57: "do not reports" should be "do not report".

Reply: Suggested modification is now in new line 57.

  1. "E. coli" should be italic.

Reply: All “E. coli” in the entire text was checked for the proper scientific writing, in italics.

Reviewer 2 Report

The authors describe here the transport of glucose in the E. coli bacterium. A great deal of work has been done to synthesize and illustrate the state of the art of knowledge on this mechanism.

Major revisions :

Lines between the different rows should be added within the 2 tables to improve readability. Improve table 2 by expanding the column on resultant phenotype. Take table 2 fit on 2 pages.

Table one should be placed at line 58; Figure 1 at line level 86.

The proposal is to put the genes in alphabetical order within table 1 within each carrier family.

Line 456 to 469: add a table to simplify and improve readability of the individual numbers.

Place figure captions on the same page as the figures.

 The perspectives of such informations and applications must be added in the corresponding paragraph.

Minor revisions :

Line 141 : b in lower case in the word bacteria.

Line 209. E coli in italicics

Line 348 : prokaryotic

Line 376 and 377: remove the ; and add commas.

Line 514 : White

Author Response

The authors describe here the transport of glucose in the E. coli bacterium. A great deal of work has been done to synthesize and illustrate the state of the art of knowledge on this mechanism.

Reply: Dear reviewer, thanks for the suggestions and comments on the first version of our contribution. Please find the point-by-point reply for each comment. All the changes in the new version of the manuscript are highlighted in the red text in the corresponding lines.

Major revisions:

Lines between the different rows should be added within the 2 tables to improve readability. Improve table 2 by expanding the column on resultant phenotype. Take table 2 fit on 2 pages.

Reply: We increased the line spacing in Table 1 and new Table 3, and as suggested, readability improved. Suggested changes were applied in new the new Table 3. The new table now fits on two pages.

Table one should be placed at line 58; Figure 1 at line level 86.

Reply: Table 1 was placed at line 58, as suggested. Regarding Figure 1, we consider the location in the original line 301 (new line 198) appropriate because the depth description of PTS starts in sections 3 and 3.1 (new line 138). The first mention of Figure 1 was made in new line 159 (in red color text). We hope you agree with this reply.

The proposal is to put the genes in alphabetical order within table 1 within each carrier family.

Reply: Suggested changes were applied to the new Table 1.

Line 456 to 469: add a table to simplify and improve readability of the individual numbers.

Reply: As suggested, the new Table 2 was included in line 460.

Place figure captions on the same page as the figures.

Reply: we reduced the size of all the figures in the new version of the manuscript to fit the corresponding captions on the same page.

The perspectives of such informations and applications must be added in the corresponding paragraph.

Reply, Section 5 (lines 527 – 561) and the new Table 3 were improved by including specific information related to the results of transport engineering, particularly the inactivation of PTS and the replacement or selection of alternative transporters for glucose transport, of course, in combination with other genetic modifications. The new information includes data on the titer and yield of desired metabolites and proteins as the simplified presentation of the impact of the inactivation of PTS:glc and other transporters on the modulation of other transporters and metabolic pathways.

Minor revisions :

Line 141 : b in lower case in the word bacteria.

Reply: The suggested change was applied in new line 145.

Line 209. E coli in italicics

Reply: The formal writing of E. coli (in italics) was revised in the entire document.

Line 348 : prokaryotic

Reply: The suggested change was applied in new line 331.

Line 376 and 377: remove the ; and add commas.

Reply: Suggested changes were applied in new line 398.

Line 514 : White

Reply: The suggested change was applied, new line 506.

Round 2

Reviewer 2 Report

Major revisions:

- table 1 : inverse column genes and transported sugar since the alphabetical order has been attributed to gene for classification

- l96 : place figure 1 here

- table 2 : separate results into 2 columns for aerobic and anaerobic conditions. change the first sentence concerning the mutant ptsG. separate informations on the first mutation by going to the line between them

- table 3 : limit line spacing to enable paragraph on stg8 to be on only 1 page. shorten the informations by giving sentences without verbs.

- add perspectives at the end of the conclusion

Author Response

Reply to the reviewer.

Dear Reviewer 2, thank you for the comments and suggestions on the revised version of our contribution. Find the reply point by point in the below lines. All the changes in the new version of the manuscript were typed in red. 

Major revisions:

- table 1 : inverse column genes and transported sugar since the alphabetical order has been attributed to gene for classification

Reply:

Table 1 was modified as requested. 

- l96 : place figure 1 here

Reply:

Figure 1 was inserted after line 96. 

- table 2 : separate results into 2 columns for aerobic and anaerobic conditions. change the first sentence concerning the mutant ptsG. separate informations on the first mutation by going to the line between them

Reply:

The new Table 2 was modified as indicated: Results were presented in two new columns, one for aerobic and the second one for anaerobic conditions. 

- table 3 : limit line spacing to enable paragraph on stg8 to be on only 1 page. shorten the informations by giving sentences without verbs.

Reply:

Table 3 was modified as requested: Line spacing was adjusted to fit the corresponding information on one page. The information in Table 3 was shortened. All revised sections are shown in red text. 

- add perspectives at the end of the conclusion

Reply:

The new Conclusion section includes perspectives on the relevance of sugar transport engineering. The latest information is written in red text.    

Round 3

Reviewer 2 Report

Minor revisions

-- in table 2 :

1- simplify the informations for the two last parental strains and avoid use of verbs

2- separate informations per line. for the first parental strain, take care to have informations to compare on the same line

-- in table 3 : separate informations per separated line as for table 2 ; avoid use of verbs 

Author Response

Minor revisions

-- in table 2 :

1- simplify the informations for the two last parental strains and avoid use of verbs

2- separate informations per line. for the first parental strain, take care to have informations to compare on the same line

-- in table 3 : separate informations per separated line as for table 2 ; avoid use of verbs 

Reply:

Dear Reviewer 2, suggestions and modifications on the content of Table 2 and Table 3 were attended. We ordered resultant phenotypic traits of the resultant mutants described in both tables in line by line reducing redundant content and verbs. Modified sections are shown in red color. 

I hope find suitable the content  of new tables.